# Inequality in Fossil Fuel Power Plants in China: A Perspective of Efficiency and Abatement Cost

Yongrok Choi [1], Yunning Ma [1,*], Yu Zhao [2] and Hyoungsuk Lee [3,*]

1   Program of Industrial Security Governance, Inha University, Incheon 22212, Republic of Korea
2   Institute of Blue and Green Development, Shandong University, Jinan 264209, China
3   Energy Environment Policy and Technology, Graduate School of Energy and Environment (KU-KIST Green School), Korea University, Seoul 02841, Republic of Korea
*   Correspondence: mayunning666@inha.edu (Y.M.); lhs2303@korea.ac.kr (H.L.)

**Abstract:** Quantifying the shadow price (SP) of $CO_2$ emissions is the key to achieving China's "double carbon" targets. Considering technology heterogeneity, this study applies stochastic frontier analysis combined with meta-frontier technology to estimate the environmental technical efficiency (ETE) and SP of $CO_2$ emissions for China's fossil fuel power plants from 2005 to 2015. This approach overcomes the lack of statistical inference and consistency of traditional methods and improves the reliability of results. The main results are as follows: (a) the average ETE of China's power plants is 0.9444, indicating that inefficient production accounts for 5.66%. The difference in efficiency between the central and local groups is significant. (b) The national average SP of $CO_2$ is 266.8 US dollars per ton, which is much higher than the carbon price in the emission trading system. This result implies the need to design a carbon trading price mechanism. (c) The distribution of SP shows obvious corporation and geographical characteristics that are closely related to the level of regional economic development. Finally, the findings provide policy implications for the improvement of the efficiency and abatement of costs of power plants and the determination of carbon prices.

**Keywords:** $CO_2$ emissions; meta-frontier stochastic frontier analysis; shadow price; China's fossil fuel power plants

## 1. Introduction

Large amounts of greenhouse gas (GHG) emissions contribute to climate change and have serious negative effects. The expansion of China's power sector has attracted enormous attention because of its contribution to GHG emissions and climate change. As shown in Figure 1, China's power sector releases approximately 46.53% of $CO_2$ emissions from fuel combustion [1], and the trend is showing obvious upward movement. Given the rapid expansion of the power sector, the Chinese government has emphasized the importance of controlling carbon emissions from power plants and has designed relevant tools, including command-and-control and carbon trading schemes [2]. The national carbon emission intensity reduction target and its delegation in each province is regarded as a command-and-control policy tool for $CO_2$ emission reduction, which was mainly implemented under the five-year plan (FYP) [3]. Another tool is a market-based regulatory strategy, which applies a tradable licensing system to reduce $CO_2$ emissions at minimal cost [4]. From the perspective of $CO_2$ emission reduction, carbon trading schemes are more effective than command-and-control policy tools [5]. Therefore, compared with command-and-control methods, the operation of the carbon trading market can more effectively enhance carbon efficiency and serve as the fastest channel to achieve China's "double carbon" targets. However, the carbon prices in the carbon trading market are problematic. The carbon price is too low to reflect the real abatement cost owing to different issues in the carbon trading market, including the lack of supervision and fuzzy quota allocation [6]. Taking carbon price as the benchmark would tremendously underestimate the Chinese

government's emission reduction budget, and such an underestimation could pose risks. Therefore, understanding actual abatement costs is necessary and even urgent. Specifically, such knowledge may contribute to the government's preparations to minimize risks and help power plants adjust the industrial structure within a reasonable budget.

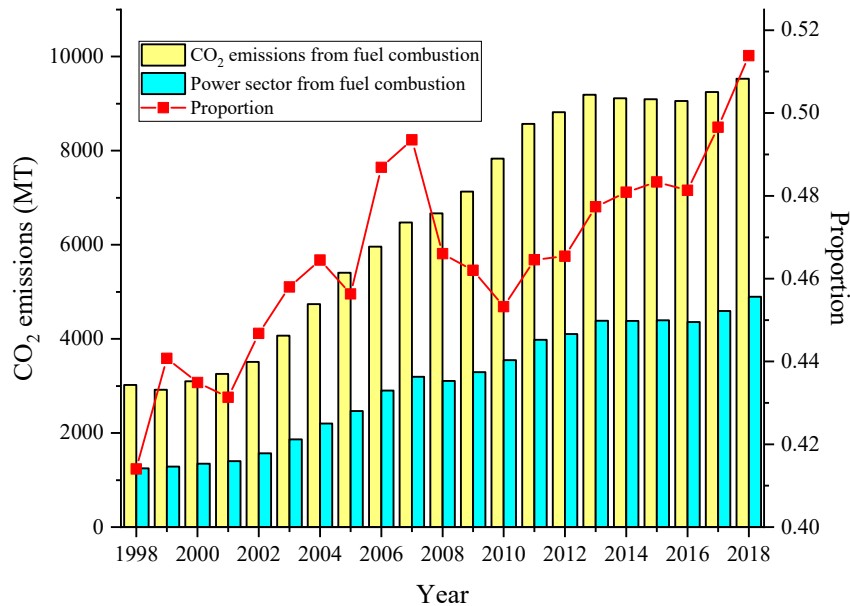

**Figure 1.** $CO_2$ emissions in power sector.

To address identified problems, scholars and practitioners have paid considerable attention to the empirical research on the power sector from the viewpoints of efficiency and abatement costs. Wei et al. [7], Du and Mao [8], Peng et al. [9], and Wei and Zhang [10] employed the parametric linear programming (PLP) approach. Zhao and Ma [11], Zhang et al. [12], Bi et al. [13], Wei et al. [14], and Xie et al. [15] used the data envelopment analysis (DEA) method, while Chen et al. [16], Wang and Jiang [17], Qi and Choi [18], Xie et al. [19], and Zhang et al. [20] applied the stochastic frontier analysis (SFA) approach. However, these existing studies assumed that power plants are consistent with technical homogeneity. Without consideration of technical heterogeneity, the estimated efficiency and abatement costs will be biased [21]. To overcome this assumption, studies have introduced DEA and PLP combined with meta-frontier analysis to the power sector in China [22–26]. Despite extensive research, we have identified a lack of investigation into the efficiency and abatement costs of China's power sector. In particular, few studies can provide meaningful statistical inferences in estimating the efficiency and abatement costs of China's power sector, and these studies depend on programming technology [27]. Thus, we use SFA as the benchmark and combine it with meta-frontier analysis to investigate the environmental technical efficiency (ETE) and shadow price (SP) of $CO_2$ in China's power sector. The derived estimator exhibits ideal statistical characteristics and is capable of making statistical inferences [28]; hence, it offsets the lack of a coherent data-generating process in pooled models [29] and the omission of estimation errors in mixed approaches [30,31]. Given the unique features of China's political and economic system, the government can directly affect the carbon market [32]. Accordingly, the consensus is for the $CO_2$ market price to be lower than the actual abatement cost; this condition may result in interest loss for regulators [33,34]. Therefore, we also compare SP and carbon market price to understand the reasons for the divergence. Such an in-depth analysis is extremely significant for policymakers as it can reveal the main defects of individual power plants and the carbon trading market. This important issue has largely been ignored in the research on the power sector, with the works of Wang et al. [35] and Xian et al. [36] representing some exceptions.

The present study uses SFA combined with meta-frontier analysis to investigate China's fossil fuel power plants from 2005–2015. By assessing power plants with different ownership types and comparing the market prices of the carbon trading market, this study answers the following two questions: first, what are the ETE and carbon abatement costs of power plants with consideration of technology heterogeneity and meaningful statistical extrapolation? Second, why is pricing in the carbon trading market lower than actual abatement costs? Currently, the power sector accounts for a relatively high proportion of $CO_2$ emissions in China and may be vital in the transformation of low-carbon technology. Therefore, this study contributes to the exploration of the carbon abatement costs of power plants. Most importantly, it provides a solid foundation for system design for the development of carbon trading schemes and the achievement of "double carbon" targets.

The remainder of this paper is organized as follows: Section 2 presents the SFA method combined with meta-analysis and relevant data. Section 3 discusses the empirical results for ETE and SP. Section 4 describes the reasons for low carbon prices in the carbon trading market. Finally, Section 5 summarizes the main conclusions and policy implications of the study.

## 2. Methodology and Data

### 2.1. Methodology

#### 2.1.1. Directional Distance Function

Färe et al. [37] defined environmental production technology while considering un-desirable outputs to measure ETE. In the current study, we consider technological hetero-geneity and undesirable outputs in estimating the ETE and SP of $CO_2$ emissions. Consider $N$ fossil fuel plants, each of which uses 3 inputs (labor ($L$), capital ($K$), and energy ($E$)) to produce 1 desirable output (electricity generation ($Y$)) and 1 undesirable output ($CO_2$ emissions ($B$)). All input–output factors constitute the environmental production possibility set $T$, which is expressed as follows:

$$T = \{(X, Y, B) : X \ can \ produce (Y, B)\} \tag{1}$$

The directional distance function (DDF) for the output is defined as follows:

$$\vec{D}(X, Y, B; g) = \max\{\beta : (X, Y + \beta g_Y, B - \beta g_B) \in T\} \tag{2}$$

where $g = (g_Y, -g_B)$. The DDF represents the simultaneous maximum expansion of the desirable output and contraction of the undesirable output for a given production technol-ogy. In general, a plant with $\vec{D}(X, Y, B; g) = 0$ is considered to have realized the production frontier. If $\vec{D}(X, Y, B; g) > 0$, then the plant still has room for efficiency improvement.

The DDF is typically shown in a quadratic function form to satisfy a non-neutral technical change [38]. The production function can be expressed as

$$
\begin{aligned}
\vec{D}(X, Y, B; g, t) = \ & \alpha_0 + \alpha_1 K + \alpha_2 L + \alpha_3 E + \beta_1 Y + \gamma_1 B \\
& + \frac{1}{2}\left[\alpha_{11}K^2 + \alpha_{12}KL + \alpha_{13}KE + \alpha_{21}LK + \alpha_{22}L^2 + \alpha_{23}LE + \alpha_{31}EK + \alpha_{32}EL + \alpha_{33}E^2\right] \\
& + \frac{1}{2}\beta_{11}Y^2 + \frac{1}{2}\gamma_{11}B^2 + \mu_{11}YB + \theta_{11}KY + \theta_{21}LY + \theta_{31}EY + \vartheta_{11}KB + \vartheta_{21}LB + \vartheta_{31}EB + \tau_1 t \\
& + \frac{1}{2}\tau_{11}t^2 + \varphi_{11}tK + \varphi_{12}tL + \varphi_{13}tE + \delta_{11}tY + \rho_{11}tB
\end{aligned}
\tag{3}
$$

Compared with the translog function, the quadratic functional form is more suitable for the translation property of the DDF. The translation property is expressed as follows:

$$\vec{D}(X, Y + \alpha g_Y, B - \alpha g_B; g) = \vec{D}(X, Y, B; g) - \alpha \tag{4}$$

This property signifies that if the vector $(X, Y, B)$ is translated into $(X, Y + \alpha g_Y, B - \alpha g_B)$, then the value of $\overrightarrow{D}(X, Y, B; g)$ is reduced by $\alpha$. The purpose of the translation property is to estimate the parameters of the DDF [39]. When combined with Equations (3) and (4), where $\alpha = B$, the quadratic DDF can be converted to:

$$
\begin{aligned}
-B = \alpha_0 + \alpha_1 K \quad &+ \alpha_2 L + \alpha_3 E + \beta_1(Y + B) + \tfrac{1}{2}\alpha_{11}K^2 + \alpha_{12}KL + \alpha_{13}KE + \tfrac{1}{2}\alpha_{22}L^2 + \alpha_{23}LE + \tfrac{1}{2}\alpha_{33}E^2 \\
&+ \tfrac{1}{2}\beta_{11}(Y + B)^2 + \theta_{11}K(Y + B) + \theta_{21}L(Y + B) + \theta_{31}E(Y + B) + \tau_1 t + \tfrac{1}{2}\tau_{11}t^2 + \varphi_{11}tK \\
&+ \varphi_{12}tL + \varphi_{13}tE + \delta_{11}t(Y + B) + v - u
\end{aligned}
\tag{5}
$$

where $u = \overrightarrow{D}(X, Y, B; g, t)$ is the technical inefficiency and $v$ is the random disturbance term with a mean value of 0 and a normal distribution. To ensure that the translation and symmetry properties hold, the parameters of the DDF must satisfy the following constraints:

$$
\begin{gathered}
\alpha_{12} = \alpha_{21}, \ \alpha_{13} = \alpha_{31}, \ \alpha_{23} = \alpha_{32} \\
\theta_{11} = \vartheta_{11}, \ \theta_{21} = \vartheta_{21}, \ \theta_{21} = \vartheta_{21}, \ \delta_{11} = \rho_{11} \\
\beta_1 - \gamma_1 = -1, \ \beta_{11} = \gamma_{11} = \mu_{11}
\end{gathered}
$$

### 2.1.2. Meta-Frontier Stochastic Frontier Analysis

The meta-frontier stochastic frontier analysis (MSFA) was first proposed by Huang et al. [28]. In general, MSFA is divided into 2 steps. In the 1st step, we estimate the group frontier using SFA. In the 2nd step, the results of the 1st step are used to construct the meta-frontier for all observations.

We divide all observations into $k(k = 1, 2, \ldots, K)$ groups. Each group has a specific production technology frontier that is not homogeneous. According to the technical levels of the different groups, we define the environmental production possibility set of each group as $T^k$.

$$
T^k = \left\{ \left( X^k, Y^k, B^k \right) : X \ can \ produce (Y, B) \right\}, k = 1, 2, \ldots, K
\tag{6}
$$

The DDF of the group is similar to that of Equation (3). Combined with the translation property, we obtain:

$$
-B^k = \overrightarrow{D}^k \left( X^k, Y^k + \alpha g_Y, B^k - \alpha g_B; g \right) + v^k - u^k
\tag{7}
$$

where $u^k = \overrightarrow{D}^k \left( X^k, Y^k, B^k; g \right)$. The ETE is written as $e^{(-u^k)}$. To obtain the ETE, parameters, and fitted values of $\overrightarrow{D}^k \left( X^k, Y^k, B^k; g \right)$, we employ the SFA model in estimating Equation (7). We can obtain the relationship between the actual and fitted values of $\overrightarrow{D}^k \left( X^k, Y^k, B^k; g \right)$ as:

$$
\hat{\overrightarrow{D}}^k \left( X^k, Y^k, B^k; g \right) = \overrightarrow{D}^k \left( X^k, Y^k, B^k; g \right) + \tilde{v}^k
\tag{8}
$$

where $\tilde{v}^k = \hat{v}^k - v^k$.

We define a meta-frontier technology possibility set as:

$$
T^m = \{ (X, Y, B) : X \ can \ produce(Y, B) \ for \ all \ plants \}
\tag{9}
$$

The meta-frontier DDF can be shown as:

$$
\overrightarrow{D}^m (X, Y, B; g) = \max\{ \beta^m : (X, Y + \beta^m g_Y, B - \beta^m g_B) \in T^m \}
\tag{10}
$$

This equation has the same meaning as Equation (6); the difference between them is that the technology frontier is heterogeneous. In addition, the meta-frontier must envelop all group frontiers in which $\overset{\rightarrow m}{D}(X,Y,B;g) \geq \overset{\rightarrow k}{D}(X,Y,B;g)$. On the basis of this relationship, we can define the technology gap difference (TGD) as follows:

$$\overset{\rightarrow m}{D}(X,Y,B;g) = \overset{\rightarrow k}{D}(X,Y,B;g) + TGD \tag{11}$$

According to Equation (11), the technology gap ratio $TGR = e^{(-TGD)}$. By substituting Equation (8) into Equation (11), we obtain:

$$\overset{\hat{\rightarrow} k}{D}\left(X^k,Y^k,B^k;g\right) = \overset{\rightarrow m}{D}(X,Y,B;g) + v^m - u^m \tag{12}$$

where $v^m = -\overset{\sim}{v^k}$, $u^m = TGD$.

Therefore, the meta-frontier environmental technical efficiency (MTE) is calculated as:

$$MTE = e^{(-u^k - TGD)} = TE \times TGR \tag{13}$$

In general, the larger the value of the TGR, the smaller the gap between the group frontier and the meta-frontier and vice versa. The higher the MTE value, the more advanced the technical level.

Figure 2 illustrates the MSFA model. Given an input–output level, that is, $(x_k, y_k)$, the distance between the projected meta-frontier points D and A includes 3 components: TGD ($TGD = \overset{\rightarrow m}{D}(.) - \overset{\rightarrow k}{D}(.)$), technical inefficiency between points A and B ($u^k$), and a random disturbance term between points B and C ($v^k$). For simplicity, we can express the relationship as:

$$D = A + v + u + TGD \tag{14}$$

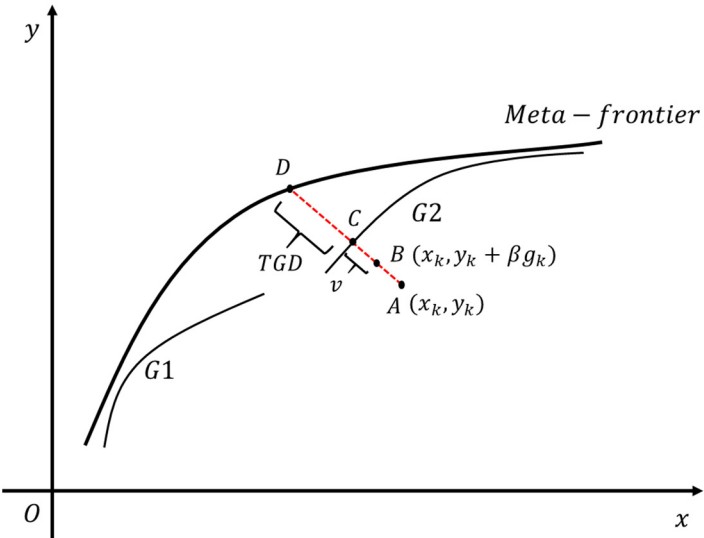

**Figure 2.** Meta-frontier stochastic frontier analysis.

### 2.1.3. Shadow Price of CO₂ Emissions

After estimating the parameters of the MSFA, we can derive the SP of the undesirable output on the basis of the DDF and its dual model [40]. The function of the SP can be expressed as:

$$q = -p \frac{\partial \vec{D}(X,Y,B;g,t)/\partial B}{\partial \vec{D}(X,Y,B;g,t)/\partial Y} \tag{15}$$

where $p$ is the price of the desirable output and $q$ is the SP of the undesirable output. Equation (15) indicates that the opportunity cost required to reduce an additional unit of undesirable output under other conditions is unchanged. By deriving Equation (3), we obtain:

$$\frac{\partial \vec{D}(X,Y,B;g,t)}{\partial Y} = \beta_1 + \beta_{11}Y + \mu_{11}B + \theta_{11}K + \theta_{21}L + \theta_{31}E + \delta_{11}t \tag{16}$$

$$\frac{\partial \vec{D}(X,Y,B;g,t)}{\partial B} = \gamma_1 + \gamma_{11}B + \mu_{11}Y + \vartheta_{11}K + \vartheta_{21}L + \vartheta_{31}E + \rho_{11}t \tag{17}$$

Combined with Equations (15)–(17), we can obtain the expansion equation of the SP as follows:

$$q = -p \frac{\gamma_1 + \gamma_{11}B + \mu_{11}Y + \vartheta_{11}K + \vartheta_{21}L + \vartheta_{31}E + \rho_{11}t}{\beta_1 + \beta_{11}Y + \mu_{11}B + \theta_{11}K + \theta_{21}L + \theta_{31}E + \delta_{11}t} \tag{18}$$

### 2.2. Data

Considering data availability, we collected the balanced panel data from 84 fossil fuel power plants in China from the period 2005–2015. A total of 924 sample observations were compiled.

#### 2.2.1. Inputs

The inputs of the DDF consisted of $L$, $K$, and $E$. We measured $L$ of each plant by the number of employees, the data for which were obtained from the Chinese Industrial Enterprises Database. The $K$ of each plant was measured by installed capacity. This information was obtained from the China Electric Power Yearbook. The data on E refer to the fuel consumption and are collected from the China Electric Power Industry Statistical Analysis.

#### 2.2.2. Outputs

The outputs include desirable and undesirable outputs. The information on Y of each plant is collected from The Compilation of Power Industry Statistical Data and is measured by electricity. The undesirable output is $CO_2$ emissions ($B$), which cannot be obtained directly. Therefore, following Wei and Zhang [10], we calculate $B$ by using the following equation:

$$CO_{2i} = \sum_{j=1}^{J} E_{ji} \times NCV_j \times CC_j \times COF_j \times \left(\frac{44}{12}\right) \tag{19}$$

where $i$ is the $i$th fossil fuel power plant, $j$ is the fuel type, $E_{ji}$ is the total consumption of each fuel type, $NCV_j$ is the total energy released by the fuel type, $CC_j$ is the carbon content, and $COF_j$ indicates the carbon oxidation factor from the Intergovernmental Panel on Climate Change [41].

#### 2.2.3. Others

When we estimate the SP of $CO_2$ emissions, we also need to use information on electricity price ($P$). Table 1 presents the descriptive statistics of all data collected from the feed-in tariff of each plant.

**Table 1.** Descriptive statistics of input and output factors.

| Variable | Definition | Unit | Obs. | Mean | Std. Dev. | Min | Max |
|---|---|---|---|---|---|---|---|
| K | Installed capacity | $10^6$ KW | 924 | 1.79 | 1.11 | 0.89 | 10.99 |
| L | Labor | Num. | 924 | 1228.19 | 799.10 | 166 | 5550 |
| E | Standard coal equivalent | $10^6$ tons | 924 | 2.61 | 0.92 | 1.22 | 7.11 |
| Y | Electricity generation | $10^9$ KWH | 924 | 0.93 | 0.51 | 0.40 | 5.67 |
| B | $CO_2$ emissions | $10^6$ tons | 924 | 7.45 | 2.61 | 3.53 | 21.10 |
| P | Price of electricity | RMB/KWH | 924 | 0.79 | 0.21 | 0.33 | 1.62 |

### 2.2.4. Classification Criteria of Data

Considering technological heterogeneity, we divided all observations into 2 groups (central and local groups) according to ownership type [42] (Table 2). The central group comprises the plants owned by the central government while the local group comprises the plants owned by the local government. In general, the central group enjoys preferential policies and financial support, whereas the local group often needs to be responsible for its profits and losses. We employ MSFA to measure and compare the ETE and SP of the 2 groups.

**Table 2.** Locations and numbers of Chinese power plants.

| Location | Central Group | Local Group |
|---|---|---|
| East | 33 | 17 |
| Central | 15 | 7 |
| West | 9 | 3 |
| Total | 57 | 27 |

Note: The central group comprises 57 power plants operated by 8 enterprises: Datang Power Group (9), Huaneng Group (14), Huadian (9), Huarun (5), Guodian (12), Guohua (3), Guotou (3), and Zhongtou (2). The local group comprises 27 local power plants.

## 3. Empirical Results

The coefficient estimates for the MSFA model are shown in Table 3 and are derived by solving Equations (7) and (12). Table 3 shows the parameter estimates for the within-group frontier, common frontier, and pooled regressions.

**Table 3.** Coefficient estimates of the MSFA model.

| Variable | Frontier | | | |
|---|---|---|---|---|
| | Central Group | Local Group | Meta | Pooled |
| $\alpha_1$ | 0.0089 | 0.0100 | 0.0180 *** | 0.0243 ** |
| | 0.0068 | 0.0266 | 0.0070 | 0.0100 |
| $\alpha_2$ | −0.5356 *** | −0.1386 * | −0.1072 *** | −0.1088 *** |
| | 0.0160 | 0.0792 | 0.0163 | 0.0234 |
| $\alpha_3$ | 0.9867 *** | 1.2087 *** | 0.9733 *** | 0.5934 *** |
| | 0.0572 | 0.2458 | 0.0596 | 0.0841 |
| $\beta_1$ | −1.0633 *** | −1.0865 *** | −0.9825 *** | −0.7694 *** |
| | 0.0339 | 0.1385 | 0.0317 | 0.0457 |
| $\alpha_{11}$ | −0.0021 | 0.0200 | −0.0045 | −0.0030 |
| | 0.0026 | 0.0167 | 0.0029 | 0.0040 |
| $\alpha_{12}$ | −0.0256 *** | 0.0150 | 0.0063 | −0.0057 |
| | 0.0080 | 0.0300 | 0.0079 | 0.0111 |
| $\alpha_{13}$ | 0.0198 | −0.4484 *** | −0.0747 *** | −0.0114 |
| | 0.0229 | 0.0743 | 0.0079 | 0.0285 |
| $\alpha_{22}$ | 0.0022 | 0.6553 *** | 0.1398 *** | 0.0612 *** |
| | 0.0117 | 0.0565 | 0.0119 | 0.0191 |

**Table 3.** *Cont.*

| Variable | Frontier | | | |
|---|---|---|---|---|
| | **Central Group** | **Local Group** | **Meta** | **Pooled** |
| $\alpha_{23}$ | 0.1504 *** | 1.7460 *** | 0.5636 *** | 0.3826 *** |
| | 0.0252 | 0.1303 | 0.0285 | 0.0428 |
| $\alpha_{33}$ | −1.4133 *** | 0.2054 | −0.9425 *** | −0.6031 *** |
| | 0.0695 | 0.2186 | 0.0601 | 0.0922 |
| $\beta_{11}$ | −0.0266 * | 1.1201 *** | 0.2942 *** | 0.1736 *** |
| | 0.0145 | 0.0694 | 0.0146 | 0.0264 |
| $\theta_{11}$ | 0.0161 | 0.2259 *** | 0.0326 ** | −0.0013 |
| | 0.0128 | 0.0454 | 0.0153 | 0.0165 |
| $\theta_{21}$ | −0.0443 *** | −1.1762 *** | −0.3690 *** | −0.2029 *** |
| | 0.0156 | 0.0863 | 0.0170 | 0.0290 |
| $\theta_{31}$ | 0.3935 *** | −1.0277 *** | −0.0037 | −0.0202 |
| | 0.0250 | 0.0968 | 0.0241 | 0.0333 |
| $\tau_1$ | 0.0034 | 0.0064 | −0.0112 *** | −0.0040 |
| | 0.0026 | 0.0062 | 0.0022 | 0.0034 |
| $\tau_{11}$ | 0.0008 * | 0.0013 * | 0.0023 *** | 0.0010 ** |
| | 0.0005 | 0.0008 | 0.0003 | 0.0004 |
| $\varphi_{11}$ | 0.0004 | −0.0046 ** | −0.0011 | −0.0009 |
| | 0.0007 | 0.0019 | 0.0008 | 0.0010 |
| $\varphi_{12}$ | 0.0018 | 0.0181 *** | 0.0395 *** | 0.0205 *** |
| | 0.0015 | 0.0059 | 0.0056 | 0.0022 |
| $\varphi_{13}$ | −0.1370 *** | −0.1393 *** | −0.1297 *** | −0.1021 *** |
| | 0.0051 | 0.0183 | 0.0056 | 0.0061 |
| $\delta_{11}$ | 0.0705 *** | 0.0644 *** | 0.0569 ** | 0.0488 *** |
| | 0.0028 | 0.0106 | 0.0033 | 0.0033 |
| Log likelihood | 1605.8979 | 634.2538 | 2140.6594 | 1920.8889 |
| Obs. | 627 | 297 | 924 | 924 |

Note: *, **, and *** represent significance levels of 10%, 5%, and 1%, respectively.

As shown in Table 3, we not only estimated the coefficients of the MSFA but also compared them with the parameters of the pooled model. On the basis of the estimations, we further calculated the ETE and technology gap and presented the results in Table 4.

**Table 4.** Estimation of ETE and technology gap.

| Group | (1) GTE | (2) TGR | (3) MTE | (4) Pooled |
|---|---|---|---|---|
| Central group | 0.9801 | 0.9664 | 0.9475 | 0.9631 |
| Local group | 0.9651 | 0.9714 | 0.9378 | 0.9729 |
| Mean | 0.9753 | 0.9680 | 0.9444 | 0.9698 |

Notes: GTE is the group-frontier environmental technical efficiency; MTE is the meta-frontier environmental technical efficiency.

Column (1) shows the results for the group-frontier environmental technical efficiency (GTE) obtained using Equation (4). Specifically, GTE is the ETE within the group frontier. The average GTE of the central group is 0.9801, whereas that of the local group is 0.9651. The GTE value of the central group is 0.015 higher than that of the local group. However, owing to the existence of different group frontiers, the efficiency values of the two groups are not comparable. Therefore, we require the introduction of meta-frontier technology. Column (2) shows the results for the TGR derived from Equation (6). The average TGR of the central group is 0.9664, whereas that of the local group is 0.9714. Compared with the central group frontier, the local group frontier is closer to the meta-frontier. Moreover, the local group obtains an advantage from technology-leading performance. From columns (1) and (2), we can further obtain column (3). The mean MTE value of the central group is 0.9475, which is 0.0097 higher than that of the local group. We believe that this result is due to fact that the policy of developing large units and suppressing small ones in China has

promoted the adjustment of the capital structure of the central group. We use the pooled model for comparison in column (4), and the average ETE is 0.9698.

The SP of $CO_2$ is calculated using Equations (10) and (11). Specifically, SP reflects the value of the electricity output that must be sacrificed to achieve a one-unit reduction in $CO_2$ emissions. Tables 5 and 6 report the SPs in the group- and meta-frontiers, respectively. As shown in Table 5, the abatement cost of an additional ton of $CO_2$ emissions for the central group is approximately 516.5 US dollars. For the local group, the SP of $CO_2$ emissions is approximately 342.1 US dollars per ton. To compare different groups, we assigned the full sample under the meta-frontier technology, as shown in Table 6. The SP of $CO_2$ emissions of the central group is 263.1 US dollars per ton, which is 11.4 US dollars lower than that of the local group. Compared with the results in Table 5, the difference in SP between the two groups based on the meta-frontier is much smaller. In reality, the abatement cost of $CO_2$ can be affected by several supply and demand factors. Consequently, the SP obtained in this study is not equal to the actual transaction price, but it can provide strong support for improving the carbon trading market.

**Table 5.** Shadow price in the group frontier (1000 US dollars/ton).

| Group | Obs. | Mean | S.D. | Min | Max |
|---|---|---|---|---|---|
| Central group | 617 | 0.5165 | 0.8516 | 0 | 8.4272 |
| Local group | 295 | 0.3421 | 0.7370 | 0 | 7.8104 |

**Table 6.** Shadow price in the meta-frontier (1000 US dollars/ton).

| Group | Obs. | Mean | S.D. | Min | Max |
|---|---|---|---|---|---|
| Central group | 627 | 0.2631 | 0.4975 | 0 | 7.0252 |
| Local group | 297 | 0.2745 | 0.3027 | 0 | 2.1846 |
| Full sample | 924 | 0.2668 | 0.4442 | 0 | 7.0252 |

Figure 3 reports the time trends of the SPs of the central and local groups. Interestingly, the SP of the local group was higher than that of the central group before 2010. After 2010, the SP of the central group exceeded that of the local group. We believe that owing to the improvement in technology, the central group needed to increase its costs to reduce one unit of $CO_2$ emissions. This result is consistent with the conclusion in Table 4. In sum, the SP of $CO_2$ generally rose after 2010. Specifically, to cope with increasingly strict environmental regulations, power plants needed to improve carbon efficiency, which further increased the opportunity cost of carbon abatement.

In addition, we conducted a heterogeneity analysis from two aspects: nine corporations and seven geographical divisions.

China's power sector is dominated by five large power generation corporations, four small power generation corporations, and other local joint ventures: Da Tang, Hua Neng, Hua Dian, Hua Run, Guo Dian, Zhong Tou, Guo Tou, and Local JV. Table 7 presents the average SPs of $CO_2$ emissions for these corporations. Among the nine corporations, Da Tang, Hua Run, Guo Hua, and Local JV exceed the average level of SP of $CO_2$ emissions by 13.12%, 34.67%, 47.34%, and 2.89%, respectively. The remaining five corporations are below the average value. The four corporations with above-average values have more advanced technology and higher efficiency than the other five. With respect to the remaining five corporations, we suggest that they increase technical investment and improve ETE.

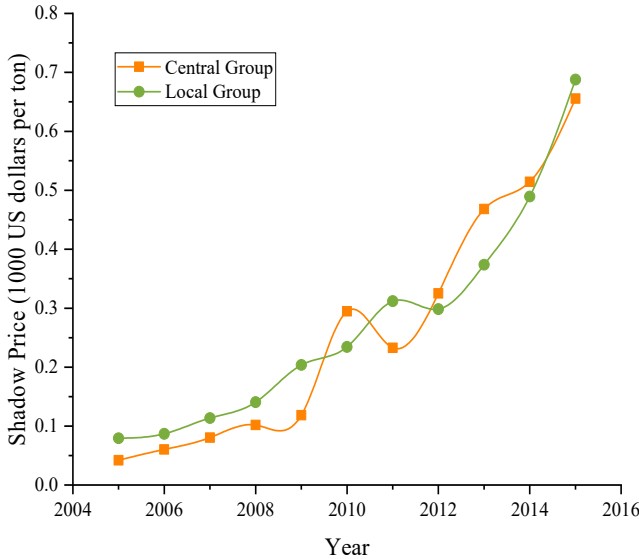

**Figure 3.** Time trend of shadow price.

**Table 7.** Average shadow prices of $CO_2$ emissions for nine corporations (1000 US dollars/ton).

| Corporation | Obs. | Mean | S.D. | Min | Max |
|---|---|---|---|---|---|
| Da Tang | 99 | 0.3018 | 0.3924 | 0.0154 | 2.5999 |
| Hua Neng | 165 | 0.2247 | 0.2598 | 0 | 1.8122 |
| Hua Dian | 99 | 0.1958 | 0.2178 | 0.0173 | 1.2114 |
| Hua Run | 55 | 0.3593 | 0.7208 | 0 | 4.6100 |
| Guo Dian | 132 | 0.2663 | 0.6320 | 0 | 7.0252 |
| Guo Hua | 44 | 0.3931 | 0.9664 | 0 | 6.4462 |
| Zhong Tou | 22 | 0.2305 | 0.2357 | 0.0238 | 0.9277 |
| Guo Tou | 11 | 0.1242 | 0.1130 | 0.0220 | 0.3735 |
| Local JV | 297 | 0.2745 | 0.3027 | 0 | 2.1846 |
| Full sample | 924 | 0.2668 | 0.4442 | 0 | 7.0252 |

We analyzed the heterogeneity of SPs from the perspective of seven geographical divisions. China has seven geographical divisions: North China, Northeast China, East China, Central China, South China, Southwest China, and Northwest China. Figure 4 shows the geographical distribution of the SP of $CO_2$ emissions. For the central group, the average SPs for the seven geographical divisions are as follows: North (665.8 US dollars/ton), Northeast (523.9 US dollars/ton), East (480.2 US dollars/ton), Central (450.6 US dollars/ton), South (552.6 US dollars/ton), Southwest (545.3 US dollars/ton), and Northwest (346.7 US dollars/ton). For the local group, the average SPs are as follows: North (464.1 US dollars/ton), Northeast (389.2 US dollars/ton), East (254.0 US dollars/ton), Central (228.0 US dollars/ton), South (338.0 US dollars/ton), and Southwest (167.1 US dollars/ton). Note that the data on Northwest China for the local group are unavailable. Intuitively, the SPs of the central and local groups increase from west to east and from north to south. This trend is directly proportional to the level of regional economic development and technical efficiency. As for the full sample, the distribution of SPs is consistent with that of the central and local groups.

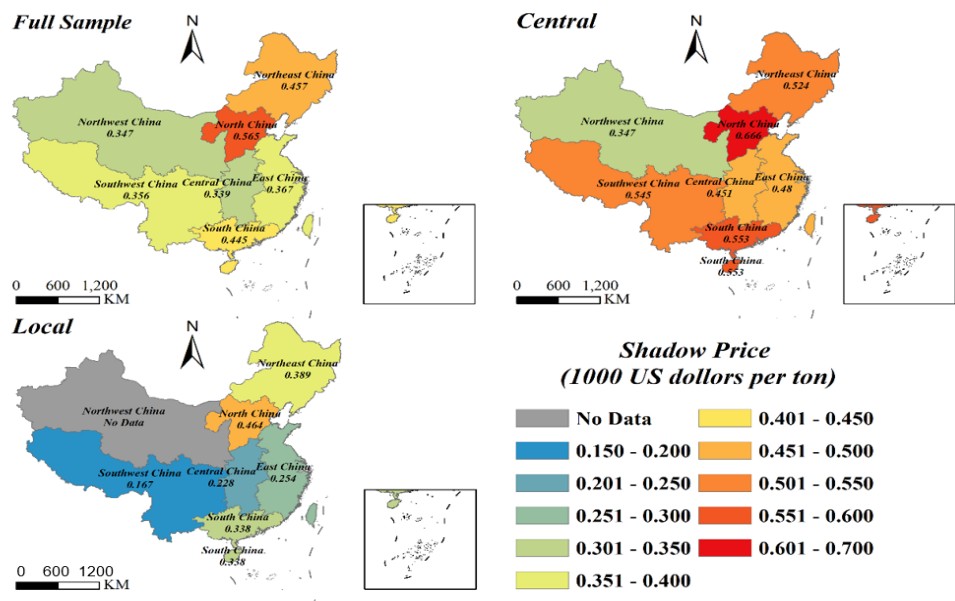

**Figure 4.** Geographical distribution of shadow prices of $CO_2$ emissions.

From the heterogeneity analysis, we find that the difference in SPs between the different regions is serious. In the carbon trading market, fossil fuel power plants whose SPs are higher than the market price are more willing to buy quotas. By contrast, plants whose prices are below the market price are more willing to sell quotas. To maximize ETE and accurately account for the carbon abatement cost, a unified carbon trading market must be established. The SP calculated from the MSFA in this study can provide an outstanding basis for the improvement of market mechanisms for emission trading systems (ETS) and the implementation of carbon peak and carbon neutrality policies.

## 4. Discussion

Our results clearly show that the SP of $CO_2$ is 266.8 US dollars per ton. The SP of $CO_2$ emissions has increased significantly since 2010 (Figure 3) because compared with those in the 11th FYP, fossil fuel power plants face stricter environmental regulations in the 12th FYP. Strict supervision could promote technological innovation in power plants and further improve abatement costs. This condition provides strong support for the Porter hypothesis [43].

However, the carbon prices of the seven carbon emission trading pilots were problematic. As shown in Figure 5, the price in the pilot market fluctuates greatly and obvious price heterogeneity exists in different pilot markets. Except for Beijing, the average carbon price in the pilot markets is lower than 50 RMB per ton. Relative to the real carbon price in this study, the price of the seven carbon emission trading pilots is too low to reflect the market supply and demand relationship. This conclusion is consistent with those of Lee [44] and Du and Mao [8].

Theoretically, carbon price is affected by the SP and demand price elasticity of power plants, as well as by the supply–demand balance of carbon licenses [45]. However, owing to the characteristics of China's political and economic system, carbon price largely depends on the relevant policies formulated by the government, such as price limits [46], carbon reserves [47], quota allocations [48], and taxes and subsidies [49]. The reason why the current carbon price is rather low is that power generation and high-emission enterprises are first launched online in the carbon trading market. The purpose of the carbon market is to introduce market-oriented means to encourage enterprises to optimize production and reduce carbon emissions, but it cannot exceed the line borne by enterprises. Meanwhile, China still has tremendous potential and demand for economic development. If China's carbon trading market is overpriced, it will face an extraordinarily heavy economic burden.

Of course, as the carbon trading market is in its infancy, the government's control over carbon prices is not sufficiently strong. Many rules, such as the quota allocation scheme and the measuring, reporting, and verification system, are key factors affecting carbon prices, but they have imperfections [9]. These deficiencies are not conducive to the scientific and reasonable emission reduction policies for power plants.

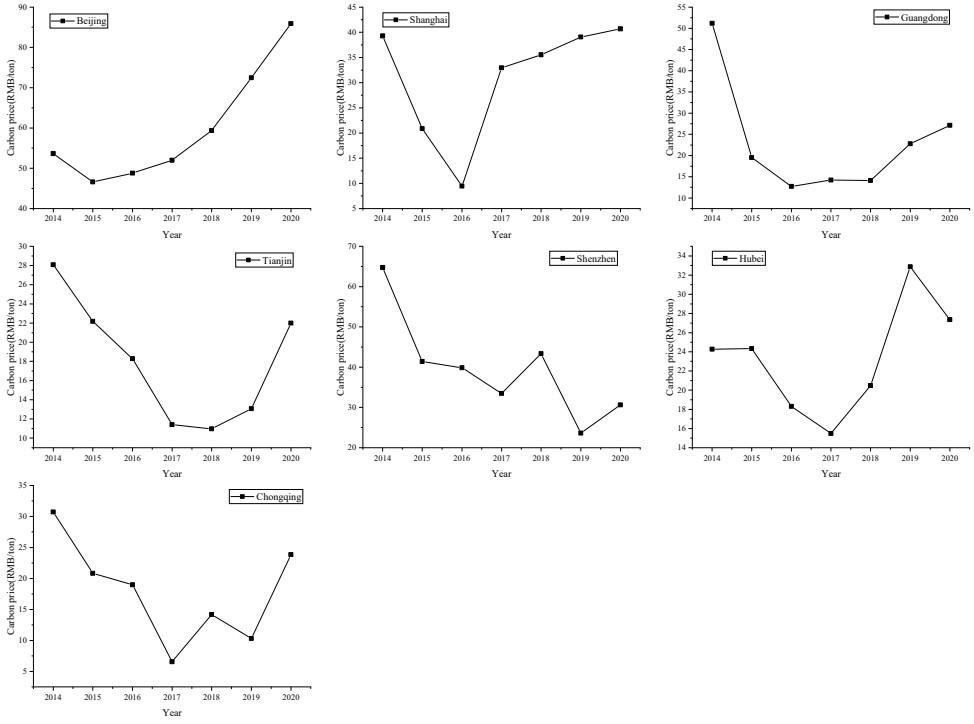

**Figure 5.** Trend of carbon prices in seven carbon emission trading pilots.

In the future, we should consider the design of a carbon market price mechanism and continue to improve the carbon price theory. Further, we should explore the influencing factors of the supply–demand relationship relative to carbon price to minimize the divergence between the market price and the actual SP and thereby promote the operational efficiency of ETS. This objective is conducive to running the market mechanism so as to shorten the distance from the "3060" target. To promote carbon efficiency and realize the carbon emission targets, the power sector can formulate specific decarbonization plans (e.g., gradual transformation from traditional energy to clean energy). In this regard, the data on energy structure and carbon oxidation factors will change accordingly. The changes will affect the quantity of carbon emissions from the power sector and may further influence the carbon efficiency ranking and carbon abatement costs of fossil fuel power plants. From our heterogeneity analysis, we find that the geographical characteristics of efficiency and abatement costs should remain unchanged. If necessary, the government can adopt different policies to promote the realization of the carbon emission targets in specific regions. For example, the Chinese government provides technical support to the west and carbon abatement subsidies to the east. Technological progress can simultaneously promote carbon efficiency and carbon abatement costs. We need to make further trade-offs between both. In the future, we need to focus on identifying the impact of carbon emission constraints to formulate optimal decisions for the decarbonization of power plants.

## 5. Conclusions and Policy Implications

Accurately estimating the cost of carbon reduction in the power sector is key to achieving the "double carbon" goal. Therefore, many studies have evaluated ETE and SP using DEA and PLP methods combined with meta-marginal analyses. However, these methods cannot present statistical inferences, thereby leading to possible in accuracies.

To overcome this statistical problem, we applied the SFA method combined with meta-marginal analysis, which is statistically inferred and consistent. In addition, we calculated the gap between the carbon price and the actual SP of $CO_2$ on the basis of this benchmark to present a field-oriented policy. The main findings of this study are summarized as follows:

First, ETE is a direct reflection of efficiency and technical level and plays an essential role in energy saving and emission reduction. The empirical analysis shows that the ETE of China's power plants is 0.9444 and that obvious differences exist between the central and local groups. In terms of efficiency decomposition, the overall efficiency of the central group is dominant. This result may stem from recapitalization and government policy support. The TGR of the local group is high, which indicates that it has gained an advantage in technology-leading performance. From the perspective of management, the result can be attributed to the improvement of the management level for the local group, which promotes the catch-up effect and narrows the gap with the meta-frontier of fossil fuel power plants.

Second, the average SP of $CO_2$ is 266.8 US dollars per ton, and the abatement cost has increased significantly since 2010. To deal with increasingly stringent environmental regulations, power plants need to improve the technical level of their carbon efficiency, which further increases the opportunity cost of carbon abatement. This finding provides strong support for the Porter hypothesis. In addition, the geographical distribution of the estimated SPs for $CO_2$ emissions shows that coastal cities and provinces (fossil fuel power plants in the eastern and southern regions) have relatively high SPs for $CO_2$ emissions owing to the strict environmental regulations and advanced production technologies. The distribution of SPs shows significant differences among corporations. For the nine corporations, Da Tang, Hua Run, Guo Hua, and Local JV exceed the average SPs by 13.12%, 34.67%, 47.34%, and 2.89%, respectively. In the future trading market, power plants and enterprises located in these areas will become potential buyers of emission quotas.

Third, we note a divergence between the actual SP and carbon price, and the implementation of the policy has an obvious impact on carbon pricing. The design of the carbon market price mechanism and carbon price theory still need to be realized.

The following policy suggestions are put forward:

First, the ETE and SP of $CO_2$ indicate the serious inequality in China's fossil fuel power plants. In other words, allocation inefficiency and technology gaps exist. Following the principle of supply–demand balance, the SP should be based on the differences in a sound carbon trading market. Hence, much room is left for carbon trading among China's fossil fuel power plants. The government should vigorously support power plants in carrying out low-carbon innovation and formulate a series of policies to promote renewable energy power generation. Furthermore, the government should provide the necessary financial resources and policies for power plants.

Second, given the heterogeneity of the geographical distribution, the government should encourage the development of a regional green economy. The results show that the SP of $CO_2$ is closely related to the level of regional economic development. Thus, the government must optimize the industrial structure and promote industrial upgrading to reduce high-energy consumption and high-emission industries. In addition, an urgent undertaking is for the Chinese government to accelerate the construction of a unified carbon trading market and bring more sectors into it.

Third, the carbon price in ETS is generally set by the Chinese government. This condition hinders efficiency maximization to a certain extent. The government should maximize the market mechanism to adjust prices and strengthen the control of GHG emissions. The government can also learn from the experiences of countries with net-zero emissions (e.g., Norway, Sweden, and France). Doing so can help the Chinese government mitigate climate change and achieve the goals of carbon peak and carbon neutrality as soon as possible.

This study has the following limitations. The data used are limited to the period of 2005–2015 and can thus be updated in future studies. Additionally, our sample cannot cover

the entire power sector, which is only a part of the entire industry. We can further discuss ETS trading in the future to provide guidance for carbon peak and neutrality policies.

**Author Contributions:** Conceptualization, Y.C.; methodology, Y.M. and H.L.; validation, Y.C. and H.L.; investigation, Y.C.; data curation, Y.M. and Y.Z.; writing—original draft preparation, Y.M. and H.L.; visualization, Y.M. and Y.Z.; funding acquisition. Y.C. All authors have read and agreed to the published version of the manuscript.

**Funding:** This research was supported by Inha University.

**Institutional Review Board Statement:** Not applicable.

**Informed Consent Statement:** Not applicable.

**Data Availability Statement:** Not applicable.

**Conflicts of Interest:** The authors declare no conflict of interest.

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
