# Peer review of "Inequality in Fossil Fuel Power Plants in China: A Perspective of Efficiency and Abatement Cost"

_sustainability, doi:10.3390/su15054365_

Round 1

Reviewer 1 Report

The authors proposed a Meta-frontier stochastic frontier analysis (MSFA) to estimate the environmental technical efficiency (ETE) and MAC of CO2 emissions for Chinas fossil fuel power plants from 2005 to 2015, which revealed the inequality of fossil fuel power plants from corporations and geographical characteristics. Regrettably, the research still has many questions and authors need to revise this paper carefully before this can be considered for the publication. There are some suggestions to improve this paper.

1. The result part, whats the meaning of using Table 2, there is no analysis for this part and the relationship with the topic isnt clearly explained.

2. The calculated shadow price are too high, much higher than the market price, even higher than the price in Western world. Have you compared it with other literature, or why the result is so high?

3. The method of MSFA, is used in this paper and should be shown in the abstract instead of SFA; Try to keep them consistent in the manuscript as sometimes the author uses SFA, or SFA combined with meta-frontier technology or Meta-frontier stochastic frontier analysis.

4. Whats the difference between shadow price (SP) and marginal abatement cost (MAC), are them same? As the authors use sometimes SP, sometimes MAC.

5. Concerning the data, 84 power plants, its better to explain how many are belong to the central group or local group, and how they distribute among corporations?

6. Concerning the sentence of “CO2 market price is lower than the actual abatement cost, which may bring an interest loss to regulators” in Line 78, why the low CO2 market price will bring an interest loss to regulators? Its not clear enough.

7. The distribution of MAC shows obvious corporations and geographical characteristics, which is inequality; but the discussion and suggestion part, the author only talk about geographical solutions, the central or local group need to be further discussed.

8. Please strengthen the explanation of your results’ significance and illustrate the innovative contribution of your work, also the logic of this paper need to be further strengthened.

9. The English expressions needs to be strengthened, some are confusing, minor word mistakes appear a lot:

Line 22: regional economic development should be the level of regional economic development;

Line 85, mate should be meta;

Line 90, there is no verb for this sentence;

Line 308, TE should be ETE?

Line 382, strength or strong?

Line 422, is or it?

Author Response

The authors proposed a Meta-frontier stochastic frontier analysis (MSFA) to estimate the environmental technical efficiency (ETE) and MAC of CO2 emissions for China’s fossil fuel power plants from 2005 to 2015, which revealed the inequality of fossil fuel power plants from corporations and geographical characteristics. Regrettably, the research still has many questions and authors need to revise this paper carefully before this can be considered for the publication. There are some suggestions to improve this paper.

-->We really appreciate your comments. Your comments were helpful to enhance quality of the paper. We reflected your concerns.

  1. The result part, what’s the meaning of using Table 2, there is no analysis for this part and the relationship with the topic isn’t clearly explained.

Response: (Because of Table 2 was added, all Table numbers were changed ex: Table2 --> Table3)

In Table 3, this table shows the parameter estimates for, respectively, within-group frontier regression, common frontier regression, and pooled regression. we derived the coefficient estimates of the MSFA model by solving equations (7) and (12) and compared them with the parameters of the ensemble model in the last column. The main significance of this figure is to show that all the correlation coefficients are reasonable and significant, and this coefficient will be used subsequently to further calculate the ETE and technology gap in Table 4

Pooled methods:Here the pooled approach is the result of regressing all the plants together without grouping

Page 8 line 265: “The Table 3 shows the parameter estimates for, respectively, within-group frontier regression, common frontier regression, and pooled regression.”

  1. The calculated shadow price are too high, much higher than the market price, even higher than the price in Western world. Have you compared it with other literature, or why the result is so high?

Response: Firstly, the market price is generally low. secondly, having referred to other literature, (Mingquan Li et al,2022) the values calculated by everyone are very high. The need for Chinese power plants to improve the technical level of carbon efficiency after 2010 in response to China's stricter environmental regulations has greatly increased the opportunity cost of carbon reduction, resulting in a calculated result that must be higher than the market price.

  1. The method of MSFA, is used in this paper and should be shown in the abstract instead of SFA; Try to keep them consistent in the manuscript as sometimes the author uses SFA, or SFA combined with meta-frontier technology or Meta-frontier stochastic frontier analysis.

Response: We replaced SFA with MSFA or SFA combined with meta-frontier technology in the article (The abstract has been revised)

  1. What’s the difference between shadow price (SP) and marginal abatement cost (MAC), are them same? As the authors use sometimes SP, sometimes MAC.

Response: Shadow prices and marginal abatement costs are essentially the same here and do not differ. To avoid confusion for the reader, we replace both marginal abatement costs with shadow prices to avoid such problems

  1. Concerning the data, 84 power plants, it’s better to explain how many are belong to the central group or local group, and how they distribute among corporations?

Response:  We added Table 2 and its detail.

  1. Concerning the sentence of “CO2 market price is lower than the actual abatement cost, which may bring an interest loss to regulators” in Line 78, why the low CO2 market price will bring an interest loss to regulators? It’s not clear enough.

Response: This is because a low carbon price will discourage and reduce the incentive of enterprises to reduce emissions, which will lead to the need for environmental regulators to devote more resources and efforts to reduce carbon emissions.

  1. The distribution of MAC shows obvious corporations and geographical characteristics, which is inequality; but the discussion and suggestion part, the author only talk about geographical solutions, the central or local group need to be further discussed.

Response:    Due to the differences in national policies and resource input, there is an objective incompatibility between the problems of central and local enterprises, and here we can only provide tentative regional solutions, while the necessary information on the problems of central and local enterprises is lacking for the time being, pending subsequent research

  1. Please strengthen the explanation of your results’ significance and illustrate the innovative contribution of your work, also the logic of this paper need to be further strengthened.

Response:  We added following paragraph at the beginning of final section to reflect your concern.

à Accurately estimating the cost of carbon reduction in the power sector is a key step toward achieving the "two-carbon" goal. Therefore, a lot of studies have evaluated ETE and MAC by DEA and PLP methods combined with meta-marginal analysis. However, these methods cannot present statistical inference which may cause lack of accuracy. To overcome this statistical problem, we applied the SFA method combined with meta-marginal analysis, which is statistically inferred and consistent. In addition, we calculated the gap between the carbon price and the actual MAC of CO2 on this benchmark to present more filed-oriented policy. Main findings of the study are summarized as follows.

  1. The English expressions needs to be strengthened, some are confusing, minor word mistakes appear a lot:

Response: We tried to reflect reviewers concern regarding English, if further English improvement needed, we are willing to use English proof service before publication.

Line 22: regional economic development should be “the level of regional economic development”;

Response: we followed your comment.

Line 85, mate should be meta.

Response: we amended it.

Line 90, there is no verb for this sentence.

Response: we amended as follow.

--> Second, why is the pricing in the carbon trading market low compared to the actual abatement costs?

Line 308, TE should be ETE?

Response: The TE here refers to the TE in Equation 13, thank you for your comment. Here, we replaced it with the ‘technology’.

Line 382, strength or strong?

Response: We modified as ‘strong’.

Line 422, is or it?

Response: We modified paragraph as follow.

--> It may stem from recapitalization and government policy support.

Reviewer 2 Report

The author presented an analysis on ETE and MAC based on empirical data, which can be helpful for rough comparison. The author made a good point on how the current carbon pricing is too low, which is helpful to drive policy change. However, I think the author needs to improve on the article before being accepted for publication. Some of the main concerns are listed below: 

1. The author needs to state clearly the limitations of the method being used and the assumptions for the method. 

2. Some of the critical parameters are very loosely defined. E.g., how is ETE defined and what the value means. It is ok to derive an equation based on empirical data to calculate it, but the author should define what the number means. Is it efficiency in avoiding environmental impact? Or is it efficiency in reducing emission? 

Some other minor issues: 

1. The first sentence is wrong. Climate change did not produce GHG emissions. GHG emissions cause climate change. 

2. Typo in L88, "Frist". 

3. Some misuse of words, e.g. strength was used instead of strong. 

Author Response

The author presented an analysis on ETE and MAC based on empirical data, which can be helpful for rough comparison. The author made a good point on how the current carbon pricing is too low, which is helpful to drive policy change. However, I think the author needs to improve on the article before being accepted for publication. Some of the main concerns are listed below: 

--> We really appreciate your comments. Your comments were helpful to enhance quality of the paper. We reflected your concerns.

  1. The author needs to state clearly the limitations of the method being used and the assumptions for the method. 

Response: Based on your comment, we added following paragraph to illustrate the limitations of the study  

--> This paper also has the following limitations. The data used in this paper for 2005-2015 can be updated in future work. In addition, our sample cannot cover the entire power sector, which is only a part of the whole industry. We can further discuss the trading of ETS in the future, which may provide guidance for carbon peak and carbon neutrality policy.

  1. Some of the critical parameters are very loosely defined. E.g., how is ETE defined and what the value means. It is ok to derive an equation based on empirical data to calculate it, but the author should define what the number means. Is it efficiency in avoiding environmental impact? Or is it efficiency in reducing emission? 

Response:    1. ETE is defined in Equation 13 as environmental technical efficiency, which in this article refers to the ratio of actual inputs to potential inputs with constant output. 2. The efficiency of avoiding environmental impacts is a broad concept, while the efficiency of reducing emissions is a narrow concept, and both concepts are consistent with this article.

Some other minor issues: 

  1. The first sentence is wrong. Climate change did not produce GHG emissions. GHG emissions cause climate change.

Response: We revised the paragraph as follow

--> Large amounts of greenhouse gas (GHG) emissions contribute to climate change and have serious negative impact.

  1. Typo in L88, "Frist". 

Response: we revised it.

  1. Some misuse of words, e.g. strength was used instead of strong. 

Response: we revised it.

Round 2

Reviewer 1 Report

The English expressions need to be further adjusted to make readers clearly understood. 

Author Response

The English expressions need to be further adjusted to make readers clearly understood. 

Response: Thanks for your professional comments. It helps to improve paper quality. As mentioned, we used English proof reading service, and attached certificate.